# When the Diagnosis of Mesothelioma Challenges Textbooks and Guidelines

**DOI:** 10.3390/jcm10112434

**Published:** 2021-05-30

**Authors:** Giulio Rossi, Fabio Davoli, Venerino Poletti, Alberto Cavazza, Filippo Lococo

**Affiliations:** 1Anatomy and Pathological Histology Unit, Infermi Hospital, 47923 Rimini, Italy; 2Operative Unit of Pathologic Anatomy, AUSL Romagna, Santa Maria delle Croci Hospital of Ravenna, 47923 Rimini, Italy; 3Thoracic Surgery Unit, Department of Thoracic Diseases, AUSL Romagna, S. Maria delle Croci Hospital, 48121 Ravenna, Italy; fabio.davoli78@gmail.com; 4Pulmonology Unit, Thoracic Diseases Department, G.B. Morgagni Hospital, 47121 Forlì, Italy; venerino.poletti@gmail.com; 5Department of Respiratory Diseases and Allergy, Aarhus University Hospital, 8200 Aarhus, Denmark; 6Department of Pathology, Arcispedale S Maria Nuova, IRCCS Reggio Emilia, 42124 Reggio Emilia, Italy; Alberto.Cavazza@ausl.re.it; 7Department of Thoracic Surgery, Faculty of Medicine, University Cattolica del Sacro Cuore, 20123 Rome, Italy; filippo.lococo@policlinicogemelli.it; 8Thoracic Surgery Unit, Fondazione Policlinico Universitario A. Gemelli IRCCS, 00168 Rome, Italy

**Keywords:** mesothelioma, pleura, histology, immunohistochemistry, BAP1, cell block, cytology

## Abstract

The diagnosis of malignant mesothelioma (MPM) does not pose difficulties when presenting with usual clinico-radiologic features and morphology. Pathology textbooks and national/international guidelines generally describe the findings of classic MPM, underlining common clinical presentation, the gold standard of sampling techniques, usual morphologic variants, immunohistochemical results of several positive and negative primary antibodies in the differential diagnosis, and the role of novel molecular markers. Nevertheless, MPM often does not follow the golden rules in routine practice, while the literature generally does not sufficiently emphasize unusual features of its manifestation. This gap may potentially create problems for patients in sustaining a difficult diagnosis of MPM in clinical practice and during legal disputes. Indeed, the guidelines accidentally tend to favor the job of lawyers and pathologists defending asbestos-producing industries against patients suffering from MPM characterized by uncommon features. The current review is aimed at underlining the wide spectrum of clinical and radiological presentation of MPM, the possibility to consistently use cytology for diagnostic intent, the aberrant immunohistochemical expression using so-called specific negative and positive primary antibodies, and finally proposing some alternative and more unbiased approaches to the diagnosis of MPM.

## 1. Introduction

Malignant pleural mesothelioma (MPM) is the most common primary tumor of the pleura and about 80% of patients with MPM have had a chronic occupational or environmental exposure to asbestos fibers [1,2,3,4,5,6]. Patients with MPM commonly consist of males aged more than 60 years at diagnosis, since the tumor requires a long latency period over 20–30 years to develop, and the prognosis is very poor [1,2,3,4,5,6]. Indeed, the median survival ranges from a few months to a couple of years, mainly depending on tumor stage, histology (epithelioid vs non-epithelioid), performance status and age at diagnosis [1,2,3,4,5,6]. Symptoms are non-specific (e.g., dyspnea and chest pain), while imaging studies commonly show unilateral pleural effusion, diffuse pleural thickening (>1 cm in thickness) with nodules, lung encasement with pulmonary volume loss, elevation of hemidiaphragm, and mediastinal shift with narrowing of the intercostal spaces [1,2,3,4,5,7]. MPM tends to extend into the thoracic structures, such as mediastinal fat, pericardium, chest wall, and mediastinal lymph nodes, while metastatic disease is rare and usually thought to involve lung parenchyma rather than distant organs [1,2,3,4,5,6,7,8].

Finally, guidelines generally recommend that the diagnosis of MPM should be made on a deep histologic sample, possibly representative of invasion of pleural soft tissues or lung parenchyma, obtained through thoracoscopy-guided biopsy [4,7,8]. Histology should be supported by a panel of positive and negative immunohistochemical markers aimed at excluding metastatic malignancies or other primary pleural neoplasms [2,3,4,5,6,8].

Needless to say, clinicians and pathologists consistently recognize MPM when the tumor presents with the aforementioned findings. However, a significant rate of MPM show a wide spectrum of clinical manifestations, unusual radiologic and morphologic appearances, aberrant or “null” phenotype at immunohistochemistry, or even combinations of all these features [9]. In addition, some patients are so fragile that they are unable to tolerate standard biopsy procedures requiring the diagnosis to be performed on non-invasive cytology at best, using a combined approach taking into consideration morphologic criteria together with ancillary investigations. In such cases, epithelioid and biphasic histology may likely be recognized, while sarcomatoid variants remain a difficult diagnostic task [8,9].

MPM often presents in advanced stage with extensive disease and when prognosis is dismal with a median overall survival < 12 months [2,3,4,5,6]. Younger age at diagnosis, epithelioid histology, and limited disease are related to better prognosis. Various nuclear grading schemes including mitotic count, necrosis, and nuclear atypia have been proposed as prognostic factors [5]. Independent from histology, a study based on The Cancer Genome Atlas (TCGA) data identified four prognostic subgroups of MPM (iClusters 1–4) using a comprehensive integrative analysis of 74 cases of MPM. The cluster with worse prognosis showed high mRNA expression of VIM, PECAM1, and TGFB1, low miR-200 family expression, low mRNA expression of mesothelin and CDKN2A homozygous deletions. PI3K-mTOR and RAS/MAPK signaling upregulation, and a significantly higher score for the Th2 cell signature [10,11,12].

Recent experiences during routine practice, external consultations or medico-legal disputations prompted us to collect some non-conventional findings of MPM that textbooks or guidelines do not sufficiently emphasize, hoping that this review article could be helpful for patients affected for some reason by challenging MPM.

## 2. Unusual Clinical Presentation

### 2.1. Mesothelioma in Patients without Definite Asbestos Exposure

Up to 20–30% of patients with malignant mesothelioma have no apparent aerogenous contact with asbestos [13,14]. At this point, the information of a significant asbestos exposure cannot be used as a diagnostic clue in the differential diagnosis between MPM and non-mesotheliomatous malignancies. Indeed, only 10% of patients exposed to asbestos seem to develop MPM. Nevertheless, knowledge of asbestos exposure is important to support the suspicion of MPM, particularly in undifferentiated/sarcomatoid malignancy showing an undefined immunohistochemical profile and prominent pleural disease at imaging studies [8,9]. Several other non-regulated, but pathogenic, mineral fibers may induce asbestosis and MPM, such as erionite or antigorite, and many others [15,16,17]. Guidelines on MPM generally underscore this issue and the lack of an updated regulation of carcinogenic mineral fibers precludes a better knowledge of carcinogenic role of fibrous minerals in promoting lung cancer and mesothelioma. Of note, MPM may occur in the ipsilateral thorax of irradiated patients for lung cancer, breast carcinoma, lymphomas or other neoplasms without any previous exposure to asbestos [18,19,20].

### 2.2. Young Age Does Not Exclude MPM

MPM typically affects elderly people, males more than females [2,3,4,5,6]. However, recent observations evidenced the occurrence of MPM in a subset of young subjects (estimated in almost 2–5% of cases presenting in patients aging less than 40 years) [21]. *Ewing Sarcoma RNA Binding Protein 1/FUS RNA Binding Protein 1-Activating Transcription Factor 1 (**EWSR1/FUS-ATF1)* gene fusions, epithelioid morphology, lack of asbestos exposure, and retained BRCA1 associated protein-1 (BAP1) expression seem to characterize this peculiar group of patients [22,23]. In another study, patients aged ≤ 35 years with MPM demonstrated a prevalent female gender, less frequent history of asbestos exposure (significant association with previous radiotherapy), no histologic differences, but higher *CDKN2A* deletion, loss of BAP1 expression, and *NF2* deletion than older patients [22].

The most commonly altered genes in MPM are *BRCA1-associated protein 1 gene (BAP1)*, *Cyclin-dependent kinase inhibitor 2A (CDKN2A,)* and *neurofibromatosis type 2 gene (NF2)* [23,24,25,26]. BAP1 gene is a major tumor suppressor gene in mesotelioma, encoding for ubiquitinase superfamily of enzymes regulating ubiquitin signalling and interfering with chromatin-associated regulating gene expression, DNA replication, and DNA repair. BAP1 loss is present in up to 60% of MPM and is caused by somatic mutations, splice alterations, gene fusions, and gene copy number alterations and is associated with improved prognosis. *CDKN2A* is a tumor suppressor gene that encodes for two distinct tumor suppressor proteins, namely p16^INK4A^ and p14^ARF^, representing critical molecules for the regulation of cell cycle pathways. *CDKN2A* deletion leads to tumor suppressor gene inactivation in many tumors including MPM. *NF2* gene encodes for merlin, a tumor suppressor protein frequently inactivated in MPM and associated with homozygous chromosomal loss and focal deletions of the 22q12 locus. Merlin regulates several intracellular events from transcription to ubiquitination through Hippo and mTOR signalling.

Possible alternative exposures to non-asbestos fibers and biological differences could explain the occurrence of MPM in young patients [27,28,29]. Recent works have highlighted the presence of *Anaplastic Lymphoma Kinase* (*ALK*) gene alterations in a subgroup of MPM [30,31,32,33], similarly to peritoneal malignant mesothelioma occurring in young patients, then representing a novel pathogenetic event and a formidable target for specific ALK inhibitors [32]. Unlike mesotheliomas in adulthood, *ALK*-rearranged mesothelioma in children and young adults appears to be similar to the aforementioned mesothelioma harboring *EWSR1/FUS-ATF1* fusions in terms of histology, but a significant predilection for peritoneum [22,27,28]. A recent analysis of MPM deaths between 1999–2015 in the United Stated evidenced that the annual number of deaths is increasing, particularly among younger populations aged <55 years, suggesting ongoing inhalation exposure to asbestos fibers and possibly other causative particles [29].

### 2.3. Atypical Clinical Onset (Paraneoplastic Syndromes)

Symptoms related to MPM are generally non-specific (e.g., cough, chest pain), and usually secondary to the presence of pleural effusion and/or diffuse pleural involvement [2,3,4,5,6]. Paraneoplastic syndromes are signs and symptoms of a tumor seen distant from the primary site or metastases. The manifestations may be endocrinologic, hematologic, gastrointestinal, renal, cutaneous, or neurologic and may occur in many different types of cancers, at varying frequencies [33]. Unlikely with other solid tumors as non-small-cell lung cancer and small-cell lung cancer, paraneoplastic disease is rarely reported in MPM. Paraneoplastic syndromes such as syndrome of inappropriate antidiuretic hormone production [34], nephrotic syndrome [35], antiphospholipid syndrome [36], polyneuropathy [37], vasculitis [38], or even production of anti-Ma2 antibodies [39,40] have been reported in MPM. Despite them usually appearing after the diagnosis of MPM, in few cases they could represent the clinical scenario of the onset, thus anticipating the diagnosis of the pleural disease.

## 3. Unusual Radiological Presentation

Imaging studies in MPM typically demonstrate unilateral effusion and diffuse thickening and nodules of the serosal surface [2,3,4,5,6,7]. However, it is poorly understood that MPM may have a broad spectrum of radiologic appearance. MPM may mimic lung cancer presenting as a solitary pulmonary localized mass or nodule (Figure 1 and Figure 2) [41,42,43,44], simulating an interstitial lung disease [45,46], showing an intrapulmonary lepidic growth with/without pneumothorax (Figure 3) [47,48] or mimicking an anterior mediastinal mass (Figure 4) [49]. In anecdotic cases, MPM debuts as acute pleural empyema [50], with monolateral [51,52], or bilateral [53] chylothorax as a consequence of a thoracic duct directly obstructing or by the presence of lymph nodal metastases in the right supraclavicular fossa. Due to its aggressive biological behavior, MPM may arise from the pleura, invading surrounding tissues of the chest wall and presenting as a laterocervical mass (Figure 5), invading bronchi (Figure 6), and regional lymph nodes (Figure 7).

Even more rarely, the clinical onset of MPM may be represented by symptoms related to the presence of distant metastases; though rare, several organs have been reported as metastatic sites, such as bone marrow [54], bone [55], breast [56], head and neck [57], intestine [58], kidney [59], liver [60], muscle [61], orbit [62] pancreas [63], oral cavity [64], skin [65,66,67], and brain [68]. Unexpectedly, different case series demonstrate that brain metastasis occurs in about 4–5% of MPM [69].

## 4. Unusual Morphologic and Immunohistochemical Features

The diagnosis of MPM in routine practice does not pose difficulties when the disease manifests with the conventional clinic-radiologic features and usual morphology of invasive growth pattern on a generous biopsy [70]. In such occurrence, the great majority of MPM look like MPM, and immunohistochemical stains have a major role for medico-legal purposes rather than for true diagnostic intent [71]. Nevertheless, the diagnosis of MPM may become particularly difficult when pathologists deal with unusual variants, since several morphologic changes have been described, such as deciduoid, clear cell, small cell, signet ring cell, pleomorphic, adenomatoid-like, and lymphohistiocytoid appearances (Figure 8 and Figure 9) [72,73,74,75,76,77,78,79,80,81]. In such challenging cases, immunohistochemical markers may reduce the diagnostic uncertainty and increase accuracy of the definitive diagnosis [81,82].

However, textbooks and guidelines on the diagnosis of MPM tend to spend a lot of words on the combined use of markers of mesothelial cell differentiation (e.g., calretinin, CK 5/6, WT-1, D2-40, HMBE1 among others) and non-mesothelial origin (e.g., Ber-Ep4, CD15, MOC31, TTF-1, CEA, claudin-4 among others), but limited mention has been made on several experiences highlighting the aberrant expression of mesothelial markers in lung tumors and vice versa [82,83,84,85,86,87,88].

Miettinen and Sarlomo-Rikala [84] investigated the expression of four so-called mesothelial markers in 596 lung carcinomas of different histology types, demonstrating calretinin expression in 67% of giant cell carcinomas, 49% of small cell carcinomas, 38% of large cell carcinomas, but rarely in adenocarcinomas. Thrombomodulin was particularly expressed in squamous carcinomas as well as cytokeratin, while 53% of adenocarcinomas immunoreacted with mesothelin. Similarly, Comin et al. [85] found thrombomodulin expression in almost all histology of lung cancer (from 71% in squamous cell carcinomas to 4% in lung adenocarcinoma), while calretinin immunoreactivity was evidenced in 44% of small cell lung carcinomas, 25% of large cell neuroendocrine carcinomas, 20% of squamous cell carcinomas, 10% of sarcomatoid carcinomas, and 4% of adenocarcinomas (4%). Of note, D2-40 was consistently expressed in 42% of squamous cell carcinomas. Overall, the results from this study indicate an aberrant reaction of mesothelioma markers in primary lung carcinomas.

Expression of cytokeratin 20 in MPM has been recently documented in epithelioid MPM (Figure 10) [89]. Most importantly, pathologists should be aware of the controversial expression of some poorly-specific primary antibody clones in MPM, such as the consistent staining of the clone SP141 of the primary antibody TTF-1 in epithelioid and sarcomatoid MPM [90,91] (Figure 11).

In particular, Klebe et al. [90] evidenced the TTF1 SP141clone nuclear expression in 8 out of 19 (42%) sarcomatoid mesotheliomas, but not with the TTF1 8G7G3/1 clone.

Again, Kushitani et al. [91] observed aberrant expression of p40 and p63 in 2 and 6 (17%) out of 36 solid epithelioid mesotheliomas, respectively, although in limited tumor areas.

Thus, unexpected immunoreactivity in MPM when using a large panel of immunostains could not mislead the pathologist from a diagnosis of MPM (Figure 12).

Of note, a careful use of immunohistochemistry is particularly required when MPM occurs concurrently with lung cancer. In a recent study from a large database (about 3800 mesotheliomas) [92], 0.5% (18 patients) (11 epithelioid, 5 biphasic and 2 sarcomatoid types) had a synchronous lung carcinoma (12 adenocarcinomas, 5 squamous cell carcinoma and 1 SCLC), and three cases had a previous history of extra-thoracic malignancy (1 testicular seminoma and bladder carcinoma and 2 with prostate carcinoma). Asbestos exposure was documented in 15 patients and those with a smoking habit in 83%.

In order to maximize the diagnostic sensitivity and specificity of immunohistochemical markers, Bernardi et al. [93] demonstrated that the coordinated use of two mesothelial negative biomarkers, namely claudin 4 and BAP1, had the best performance in differential diagnosis between epithelioid or biphasic mesothelioma and metastatic carcinomas. The expression of claudin-4 alone excluded MPM and double negativity was evident in all MPM. BAP1-positive/claudin-4-negative expression was observed almost only in MPM with epithelioid features, while a single case of BAP1/claudin-4 negative metastatic squamous cell carcinoma from the anal region was observed. This study summarized previous experiences singly investigating claudin-4 and BAP1 [94,95,96,97,98], and was performed on cell blocks and the corresponding pleural biopsies. In a practical algorithm [93], BAP1 and claudin-4 seems to represent the best panel for differentiating MPM with epithelioid features and metastatic carcinoma either on cytology or biopsy. More recently, cytoplasmic loss of methylthioadenosine phosphorylase (MTAP) has emerged as a specific marker of malignancy in mesothelial proliferations [99,100,101]. Complete loss of MTAP is a reliable marker of malignancy, correlating to homozygous deletion of CDKN2A. The sensitivity of MTAP is similar to BAP1, but the combined use of these two markers increases the overall sensitivity in recognizing MPM. In a small subset of cases, the partial loss of MTAP expression and the cytoplasmic localization of the immunohistochemical signal may result in difficult interpretation [99,100,101].

Non-epithelioid histology is frequently associated with a controversial or “null” immunoprofile for mesothelial markers [8,9,81,102,103]. Indeed, sarcomatoid MPM and its desmoplastic form often lack any expression with mesothelial markers, retaining some reactivity only with pan-cytokeratins (Figure 13). In these cases, clinicians, pathologists, and lawyers too should consider that results for non-mesothelial markers (e.g., claudin-4 and CEA) do have a greater role than positive stains for primary antibodies of mesothelial differentiation. A positive immunoreactivity for highly-specific non-mesothelial markers, in particular with claudin-4, tends to exclude MPM, although pulmonary sarcomatoid carcinomas and some sarcomas are negative for claudin-4 [9,102]. Sometimes it is impossible to differentiate sarcomatoid MPM from sarcomatoid carcinoma if immunocytochemistry for cytokeratin is positive and mesothelial markers are negative. In these cases, a careful correlation of histopathological findings with clinical and imaging information is mandatory [103,104,105]. This represents the most frequent condition during medico-legal disputes, and textbooks/guidelines do not sufficiently cover this issue, unintentionally creating diagnostic controversies. According to the Italian guidelines of the Medical Oncology Society (AIOM) [106], the careful attention posed to firmly confirm a diagnosis of MPM should also be expected to rule out MPM. A simplistic diagnosis of non-mesotheliomatous malignancy not otherwise specified should not be accepted, necessarily requiring the same careful attention posed in the diagnosis of MPM and then a final, precise pathologic definition of the neoplasm involving the pleura.

Another important and unusual morphologic presentation of malignant mesothelial proliferations is the in-situ mesothelioma (MIS). This concept has been recently reintroduced by a consensus of expert pathologists. MIS is currently considered a precursor to epithelioid MPM, raising the possibility for its role in different molecular pathways for various histology types of MPM. The diagnosis requires consistent demonstration of malignancy (e.g., loss of nuclear labelling for BAP1) in flat, mesothelial proliferations in the absence of invasion defining MIS and may be the spy of an invasive MPM in the non-sampled adjacent areas. Such a diagnosis should be posed in cases with chronic pleural effusion and no evidence of tumor growth at imaging and thoracoscopy. At histology, MIS is a proliferation of single layers of atypical mesothelial cells on pleural surfaces without invasive features after complete analysis of the specimen with serial sections. A multidisciplinary discussion of MIS is then recommended [107,108].

## 5. Diagnosis of Mesothelioma on Cytology

Another difficult task in the diagnosis of MPM is represented by the availability of a cytologic sample or a superficial biopsy lacking obvious invasion only.

Although recent guidelines from for the International Mesothelioma Interest Group (IMIG) and the International Academy of Cytology (IAC) [109] robustly evidenced the possibility to reach a diagnosis of MPM (mainly epithelioid and biphasic forms) by non-invasive diagnostic modalities using conventional cytology and/or cell blocks supported by immunostains (Figure 14), this issue still may represent a hurdle during medico-legal litigations.

Among cytological features indicating MPM in pleural effusions, the extent of mesothelial proliferation, the presence of papillary structures, scalloped borders of cell clumps, intercellular windows, variation of cytoplasmic staining and its density, and low nuclear-to-cytoplasmic ratios. Since some of these cytomorphological findings may be observed at some level in reactive mesothelial cells, the differential diagnosis of mesothelial proliferations may be very difficult or even impossible in cytology, underscoring the importance of ancillary techniques to reach a definitive diagnosis.

Nevertheless, presence of cellular atypical features, demonstration of mesothelial differentiation and of some genetic alterations including cyclin-dependent kinase-inhibitor 2A/p16 and neurofibromatosis type 2 (NF2) or lack of BRCA1-associated protein 1 (BAP1), alone or in combination, may effectively distinguish MPM from metastasis in samples, from pleural effusions (Figure 15).

BAP1 immunostaining is by far the most important biomarker in the distinction between benign and malignant mesothelial proliferations with a complete specificity, since BAP1 loss in mesothelial cells is absolutely indicative of malignancy [93,94,95,109,110]. In addition, BAP1 staining loss is particularly helpful in confirming MPM from metastatic lung or extrapulmonary carcinomas (usually showing positive nuclear reactivity) [110,111].

## 6. Heterogeneity of Mesothelioma

Heterogeneity in MPM is a poorly-debated issue, but represents an important finding either in diagnosis, prognosis, or therapeutic prediction (Figure 16). In fact, distinction of epithelioid versus non-epithelioid morphology is a well-recognized key factor in MPM management, but determination of histologic subtype by pleural biopsy proved correct in 80% of patients, as reported by Bueno et al. [112]. In their experience, most patients (174 out of 192) with epithelioid MPM showed a concordant diagnosis between pleural biopsy and resection specimens. By contrast, 45 out of 103 (44%) with non-epithelioid MPM on surgical specimen were originally misdiagnosed as epithelioid type, demonstrating a specificity of 56%. This is clearly due to the difference in size between biopsy and surgical samples, limiting the examined tumor area in small biopsy, as also recently suggested by Shulte et al. [113].

Not surprisingly, MPM shows different molecular setups and distinct allelic mutational frequencies in various sampled areas of the tumor, then demonstrating the presence of tumor subclones and high intra-tumoral heterogeneity rate [114].

Of note, non-homogenous deletions of CDKN2A in MPM is another diagnostic tool underlying the intra-tumoral polyclonal origin of mesothelial tumor cells comprising various genetic subclones [115].

Supporting this view, Comertpay et al. [115] used the HUMARA (Human Androgen Receptor) assay to investigate 16 biopsies from 14 women with MPM. Distinct methylated HUMARA alleles were observed in 14 out of the 15 informative samples supporting a polyclonal origin of MPM. The high degree of intra-tumoral heterogeneity detected even in MPM is somehow supported by the frequent patchy expression of immunohistochemical markers used for diagnosis.

Recently, a deep convolutional neural network (MesoNet) technology specifically customized to analyze whole-slides images of MPM evidenced the presence of heterogeneous components either in tumor areas (sarcomatoid features) or stroma (inflammation) more accurately than pathologists, also significantly correlating with patients’ survival [116]. This deep learning algorithm highlighted the great heterogeneity of various parts of MPM. Based on this artificial intelligence approach, expert thoracic pathologists proposed another morphologic subtype of MPM more recently, namely transitional histology, in the attempt to identify a subgroup of MPM with dismal prognosis and challenging morphology between epithelioid and sarcomatoid feature [116]. This transitional form is characterized by solid, sheet-like growth of strikingly cohesive elongated epithelioid cells with well-defined cell borders tending to spindle changes [117].

The heterogeneity MPM is then observed in morphologic, phenotypic, and genetic grounds, possibly representing another barrier for the correct diagnostic recognition and successful treatment of MPM [22,118,119,120,121,122].

## 7. Pathologic Findings in Litigations

Apart from the demonstration of asbestos exposure causing MPM (e.g., occupational exposure or carcinogenic role of chrysotile or other fibrogenic fibers) [123,124,125], there are several problems leading to legal disputes in the diagnosis of MPM crucially involving pathologists. As demonstrated by van der Bij et al. [126], 6% (97 out of 1498 patients) of MPM patients had no pathologic diagnosis because of an uncertain diagnosis or inadequate/unavailable tumor samples.

Clinical diagnosis of MPM is not accepted worldwide and a prompt biopsy or cytologic sample with cell block preparation may consistently facilitate a definitive diagnosis.

Klebe et al. [127] demonstrated that ‘expert’ second opinion can differ significantly from the original diagnosis in 53% of cases. This is particularly true when faced with sarcomatoid MPM for which immunohistochemistry is of less value than epithelial or biphasic MPM [9,101,103].

In their experience, Klebe et al. [127] highlighted several findings justifying the diagnostic discrepancies between referring to laboratory and “expert” diagnosis, including the adequacy and orientation of the biopsy, the identification of tumor invasion, the discrimination of fibrous pleuritis from sarcomatoid mesothelioma, the interpretation of discordant immunostains, and recognition of MPM from other mimicking malignancies (e.g., sarcomatoid carcinoma, synovial sarcoma, epithelioid hemangioendothelioma) [128].

In addition, at least in our own personal experience, one of the most critical issues in generating doubt on a diagnosis of MPM is secondary to the dogmatic role of immunostains stated in guidelines, not sufficiently underlying the possible controversial issues related to immunohistochemical results in MPM and non-mesothelioma. In this setting, the negativity of some mesothelial markers and/or the positivity of non-mesothelial markers are sneakily used to exclude MPM, either in sarcomatoid and epithelioid histology.

## 8. Key Messages

(1)All MPM are potentially medicolegal cases. Pathologists and lawyers should consider the possibility of all unusual presentations of MPM during diagnostic routine practice and medicolegal litigations.(2)Unusual presentation of MPM may occur on a clinical level, imaging studies, morphology, immunostains, and even in various combinations of the aforementioned conditions.(3)Diagnosis of MPM with an epithelioid component may be robustly performed on cytology/cell blocks and should not be rejected as priori in absence of a biopsy, requiring a careful correlation among clinic-radiologic findings, cellular atypia, demonstration of mesothelial cell differentiation, and possibly genetic alteration (e.g., CDKN2A or BAP1 loss).(4)Heterogeneity is an underscored aspect in MPM that will have a deep impact in the next future, involving diagnosis, prognosis, and therapeutic approaches.

## Figures and Tables

**Figure 1 jcm-10-02434-f001:**
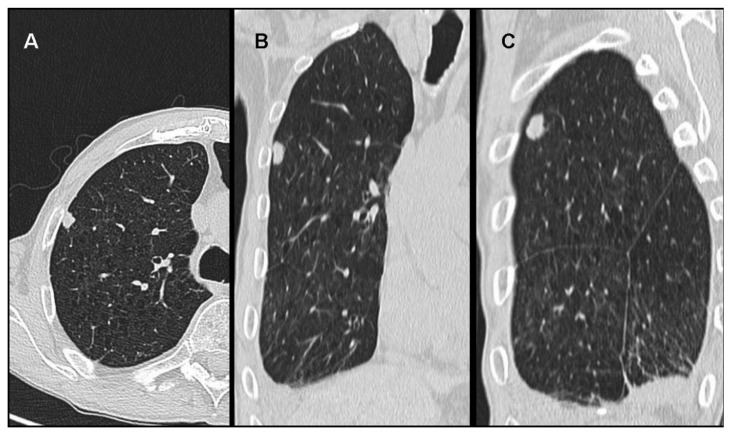
Malignant mesothelioma presenting as a solitary pulmonary nodule of the right upper lobe at the CT Scan in axial (**A**), frontal (**B**) and sagittal (**C**) views.

**Figure 2 jcm-10-02434-f002:**
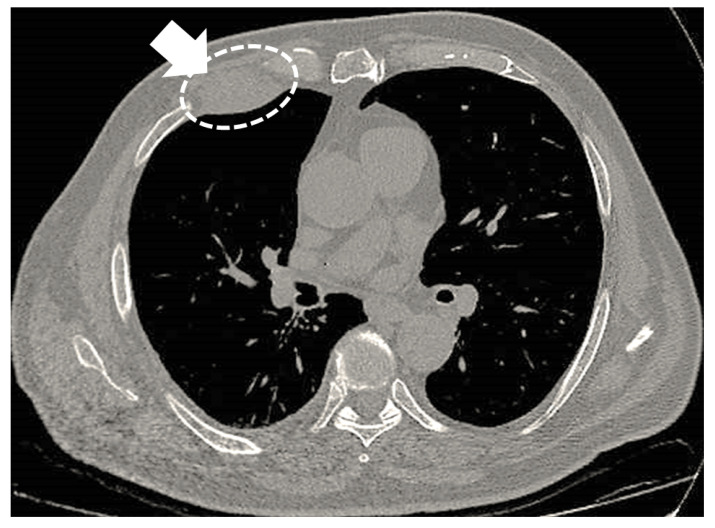
Malignant mesothelioma appearing as a localized mass of the right thoracic wall (arrow and circle).

**Figure 3 jcm-10-02434-f003:**
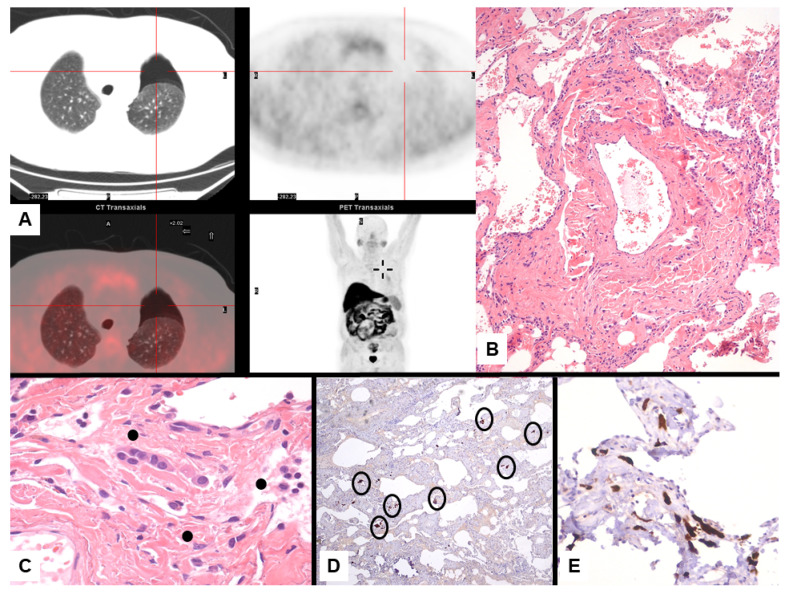
Mesothelioma presenting with left pneumothorax at imaging study with PET/CT scan without evidence of pleural thickening (**A**). Lung resection of the left apex shows interstitial fibrosis with tiny, scattered, bland-looking proliferation of epithelioid cells (**B**) partially lining alveoli and merging in the fibrosis (**C**, dots). Calretinin staining highlights the mesothelial differentiation of epithelioid cell nests randomly scattered through the lung parenchyma (**D**, circles) lining the alveolar spaces (**E**).

**Figure 4 jcm-10-02434-f004:**
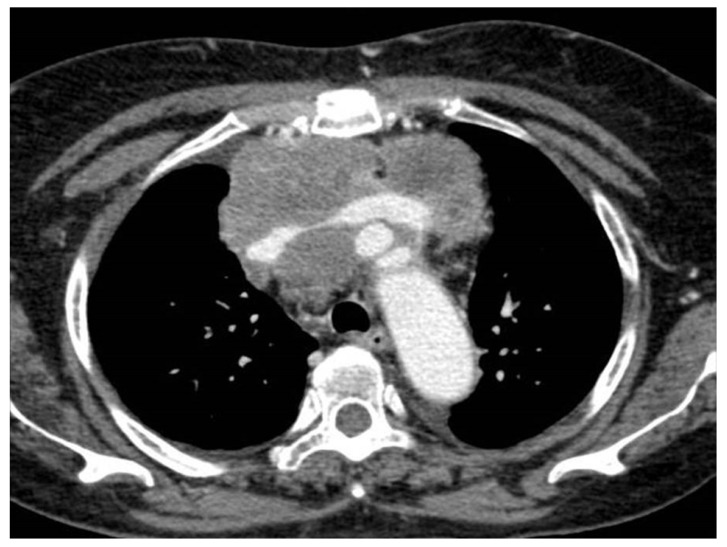
Malignant pleural mesothelioma mimicking a huge anterior mediastinal mass with compression of the brachiocephalic venous trunk at the CT scan.

**Figure 5 jcm-10-02434-f005:**
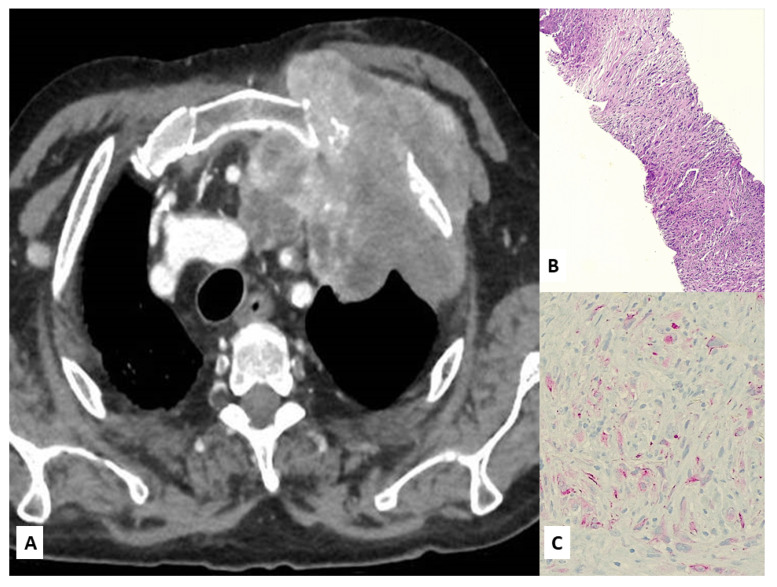
Chest CT scan showing a laterocervical mass arising from the superior sulcus of the left lung (**A**). A surgical biopsy revealed a sarcomatoid mesothelioma (**B**) focally expressing D2-40 (**C**).

**Figure 6 jcm-10-02434-f006:**
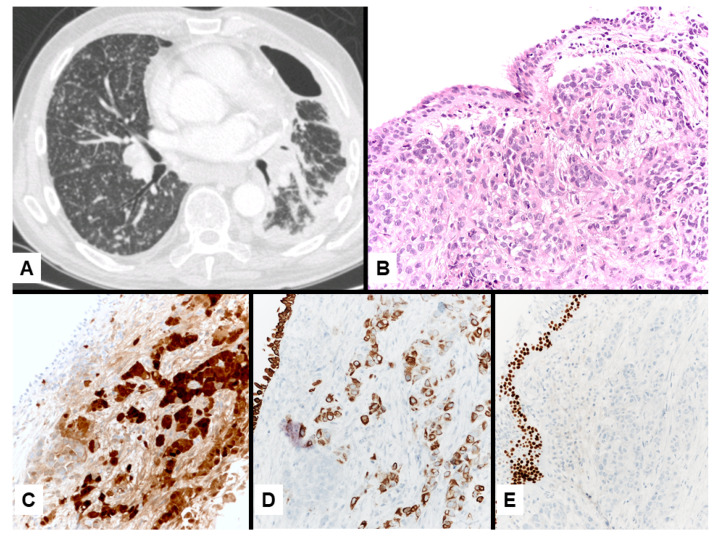
Epithelioid malignant mesothelioma of the left lung (**A**) diagnosed on a bronchial biopsy (**B**) infiltrated by a monomorphic epithelioid growth expressing calretinin (**C**) and CK5/6 (**D**), but not for p63 (**E**).

**Figure 7 jcm-10-02434-f007:**
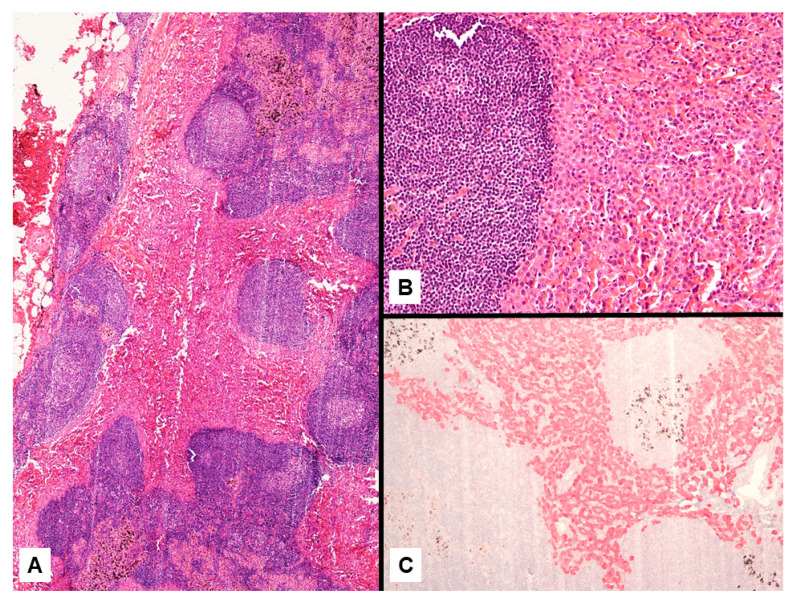
Epithelioid mesothelioma presenting as metastatic lymph node of the mediastinum (**A**) showing a solid proliferation of monomorphic epithelioid cells (**B**) strongly expressing calretinin (**C**).

**Figure 8 jcm-10-02434-f008:**
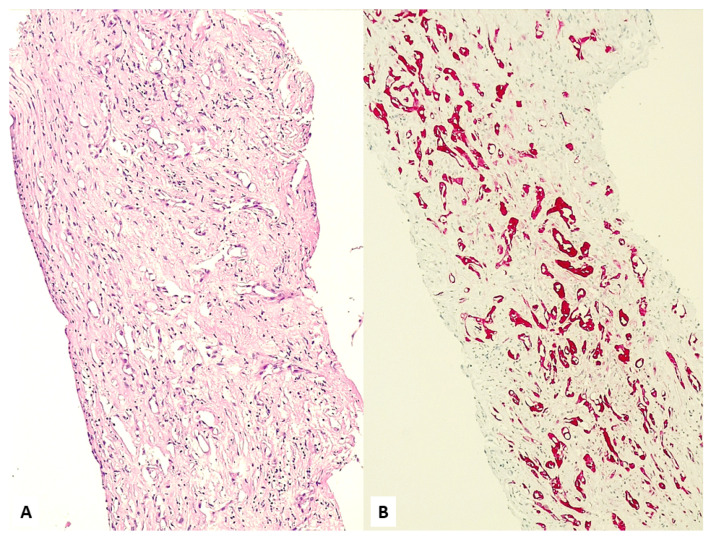
An unusual case of epithelioid mesothelioma with bland-looking adenomatoid-like spaces (**A**) diffusely expressing calretinin (**B**).

**Figure 9 jcm-10-02434-f009:**
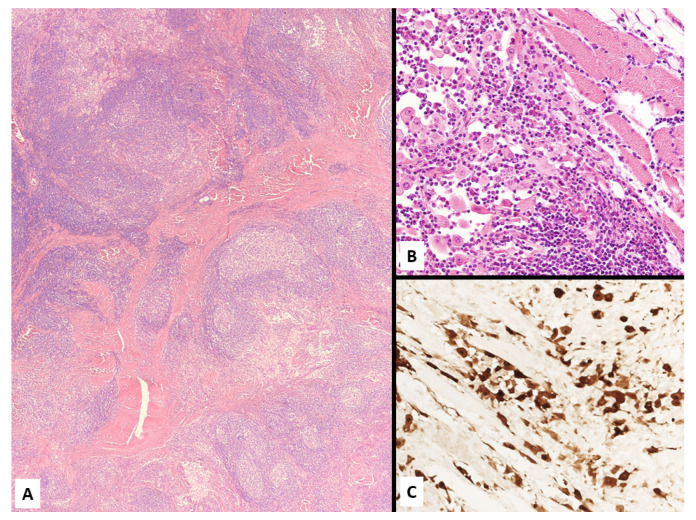
Lymphohistiocytoid mesothelioma consisting of a heterogeneous growth of mixed inflammatory cells (**A**, mainly lymphocytes and plasma cells) dissected by fibrotic tissue and discohesive malignant epithelioid mesothelial cells (**B**) expressing calretinin (**C**).

**Figure 10 jcm-10-02434-f010:**
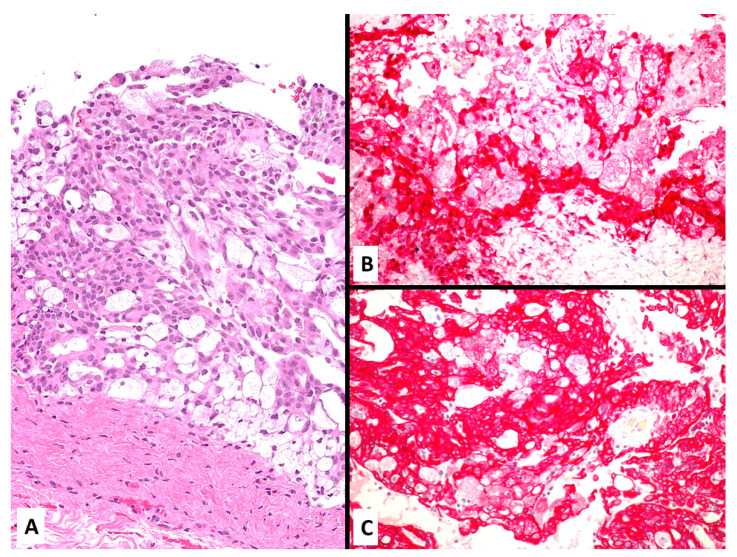
Clear cell malignant mesothelioma with proliferation of large-sized cells with foamy cytoplasm (**A**) strongly staining with calretinin (**B**) aberrantly expressing CK20 (**C**).

**Figure 11 jcm-10-02434-f011:**
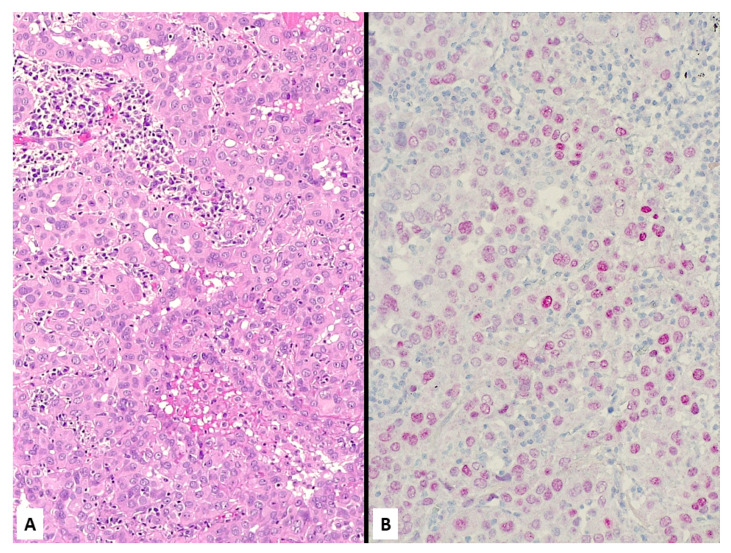
Epithelioid malignant mesothelioma (**A**) aberrantly showing nuclear expression of TTF-1 (clone SP141). (**B**).

**Figure 12 jcm-10-02434-f012:**
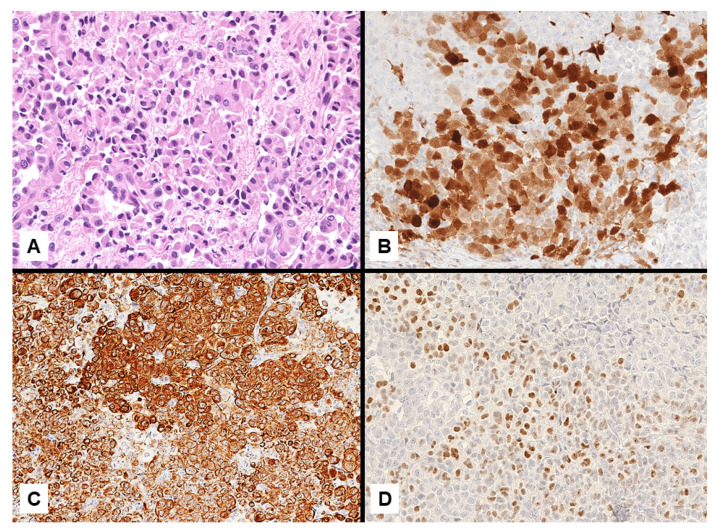
Epithelioid malignant mesothelioma (**A**) consistently staining calretinin (**B**) and CK5/6 (**C**), but unconventionally displaying nuclear expression of CDX2 (**D**).

**Figure 13 jcm-10-02434-f013:**
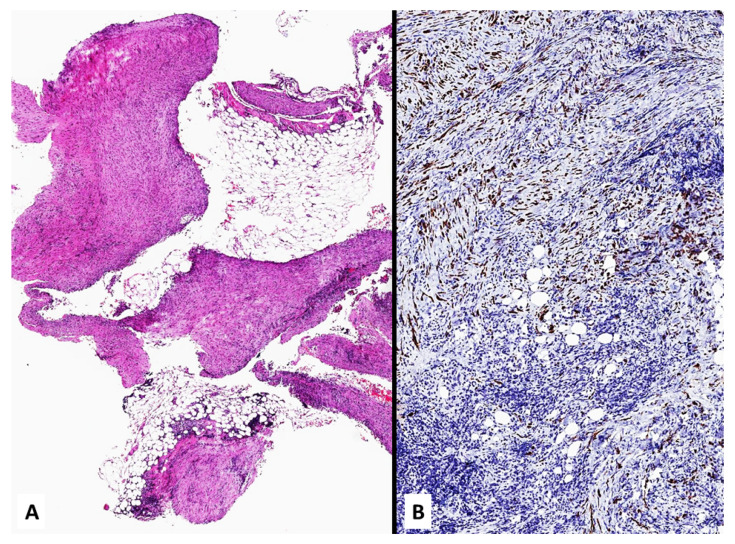
Numerous fragments of parietal pleura with a dense proliferation of spindled cells (**A**) invading soft tissue and expressing pan-cytokeratin (**B**).

**Figure 14 jcm-10-02434-f014:**
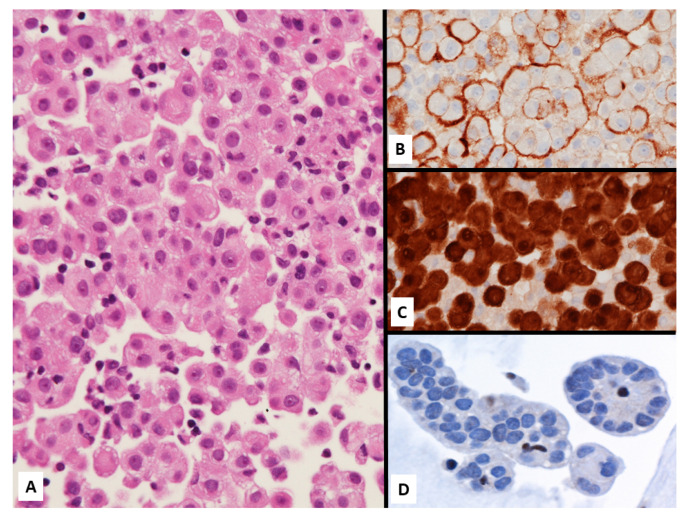
Epithelioid malignant mesothelioma on cell-block from pleural effusion showing numerous atypical cells focally forming papillary structures (**A**) expressing D2-40 (**B**) and calretinin (**C**) with loss of BAP1 staining (**D**, note some normally positive inflammatory cells among malignant cells).

**Figure 15 jcm-10-02434-f015:**
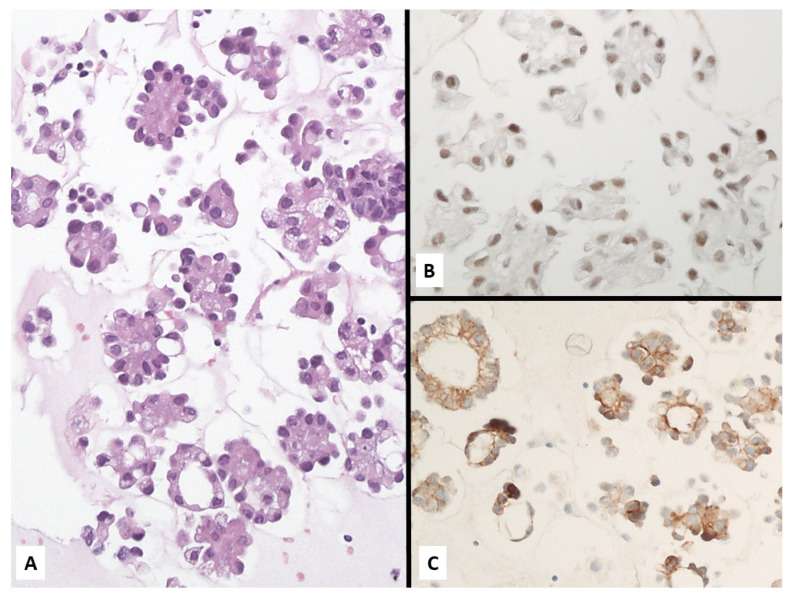
Metastatic breast carcinoma showing malignant epithelioid cells arranged in tiny nests (**A**) coordinately expressing BAP1 (**B**) and claudin-4 (**C**).

**Figure 16 jcm-10-02434-f016:**
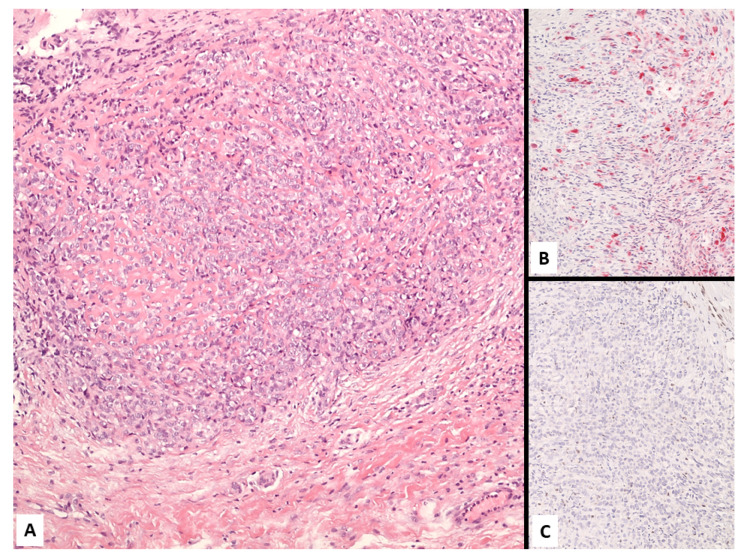
A poorly-differentiated malignant mesothelioma consisting of monomorphic small-sized epithelioid cells (**A**) showing heterogeneous expression of calretinin in scattered tumor cells (**B**). Negative staining with BAP1 supported the diagnosis of malignant mesothelioma (**C**).

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
