# Peer review of "When the Diagnosis of Mesothelioma Challenges Textbooks and Guidelines"

_jcm, 2021, doi:10.3390/jcm10112434_

Round 1

Reviewer 1 Report

Rossi et al., have submitted a manuscript titled “When the Diagnosis of Mesothelioma Challenges Textbooks and Guidelines”

The submitted manuscript details clinical aspects of Malignant pleural mesothelioma (MPM) diagnosis and challenges associated with insufficient literature in the context of unusual manifest features.

The authors establish a detailed framework of unusual clinical presentations, radiological findings, morphological and immunohistochemical features. Atypical MPM incidence, from non-asbestos exposure, gene fusions, radiological, immunohistochemical, morphological and cytological characteristics and tumor heterogeneity are also described.

Deficiencies:

  • Gene names as abbreviations are presented throughout the manuscript, however gene annotations (function and roles) are not provided, this has reduced the quality of the work.
  • Classical prognostic and diagnostic tools such as survival curves based on histology, clinical gold standard employing Mesomark and combinations of serum markers to improve diagnostic accuracy are necessary background information that must be included to convey a present clinical diagnostic practices.
  • The mechanistic link between loss of CDKN2A, NF2, BAP1 and their possible roles in promoting MPM oncogenesis should be addressed. This serves as a possible alternate mechanism in contrast to asbestos exposure leading to neoplastic development.
  • There is an apparent lack of address on medicolegal issues, the abstract (line 8-9) mention is not supported with any medico-legal instances but only touches on it very briefly (page 2, paragraph 3). The mention of medico-legal appears again on page 19, key messages point 1. Please integrate how medico-legal cases circumvent or benefit from classical text book definitions and what must be a proper approach to counter the existing flaw.
  • Non-asbestos fibrous material (reference 16), and the various atypical variables are loosely connecting this complex issue of nonasbestos, non-fibrous materials. Please integrate the following “Asbestos is not just asbestos: an unrecognised health hazard, Francine Baumann , Jean-Paul Ambrosi, Michele Carbone. Lancet Oncol. 2013 Jun;14(7):576-8. doi: 10.1016/S1470-2045(13)70257-2.” Perhaps a few more articles addressing this issue should further convince the clinical practitioners of the inadequacy of classically oriented and guidelines based literature on MPM diagnosis.

Minor spelling and wording errors:

  • Page 2, last paragraph, second last line, please correct neoplastic disease + is rarely reported in MPM. Please remove the + sign.
  • Page 10, 2nd paragraph, 2nd line, Most important, change to Most importantly
  • Page 10, 6th paragraph, 1st line, Of note, a careful use of immunohistochemistry in particularly required since MPM may occur concurrently with lung. Please rewrite this sentence.

Author Response

We thank the reviewer for the comments. Hope to have improved the review accordingly.

Best regards

Giulio Rossi

Reviewer 2 Report

‘When the Diagnosis of Mesothelioma Challenges Textbooks and Guidelines’ by Rossi G et al has been submitted to Journal of Clinical Medicine as Review article. It is a review for diagnosis of mesothelioma with clinico-radiologic features and morphology. It is well written and almost covers the problem for the diagnosis of mesothelioma. However, there are some inaccurate descriptions about the immunohistochemistry for the diagnosis of mesothelioma which will mislead the readers. As the authors write in Introduction, mesothelioma can be diagnosed on cytology based on morphologic criteria and ancillary tests. Although the authors wrote that one of their aims of this review article is to show the possibility of use of cytology for diagnostic intent in Abstract, there is no description about morphological criteria for the diagnosis of mesothelioma on cytology in the text.

Page 2. Unusual clinical presentation

The authors should describe anaplastic lymphoma kinase (ALK) gene rearrangement in children and young adults with peritoneal mesothelioma.

Hung YP et al. JAMA Oncol. 2018 Feb 1;4(2):235-238.

Mian I, et al. J Thorac Oncol. 2020 Mar;15(3):457-461.

Page 10, 7th paragraph.

The description about BAP1-positive/Claudin-4 is not true. BAP1-positive/Claudin-4 negative expression is observed in malignant mesothelioma or in reactive mesothelial cells. The authors should read the manuscript below carefully and revise this description.

Bernardi L, et al. Cancer Cytopathol. 2020 Oct 12. doi: 10.1002/cncy.22368.

Page 10, 8th paragraph.

The authors wrote that ‘negative staining with claudin-4 with/without expression with mesothelial markers robustly favor MPM in non-epithelioid histology [9,57]. It is not true, because most of sarcomadoid carcinomas of the lung are negative for claudin-4 and sarcomas also are also negative for claudin-4. The authors should read the manuscript below and revise this description.

Ordonez NG. Am J Clin Pathol 2013;139:611-619.

It is difficult to distinguish sarcomatoid malignant mesotheliomas from spindle cell and pleomorphic carcinomas of the lung. The authors should mention that it is impossible to differentiate these two if immunocytochemistry for cytokeratin is positive and mesothelial markers are negative and it is needed to correlate the histopathological findings with clinical and imaging information

Marchevsky AM et al. Human Pathology (2017) 67, 160–168.

Page 16. Diagnosis of mesothelioma on cytology

They should describe the morphology of mesothelioma cells in effusion briefly. The authors should write that the differential diagnosis of mesothelioma in effusions includes reactive mesothelial cells and metastatic carcinomas.

Although the authors wrote about BAP1 immunostaining and FISH analysis of CDKN2A for the distinction between benign and malignant mesothelioma, they should also describe methylthioadenosine phosphorylase (MTAP) immunohistochemistry for this differentiation. FISH analysis of CDKN2A is popular for this differentiation and MTAP immunohistochemistry is now thought to be useful surrogate for FISH analysis of CDKN2A. The authors should cite the manuscripts below.

Hida T, et al. Lung Cancer. 2017;104:98-105.

Berg KB, et al. Arch Pathol Lab Med. 2018;142(12):1549-1553.

The authors wrote possibility of cytological diagnosis of mesothelioma using genetic alterations of the atypical cells in effusion. However, it is not possible to confirm that an individual patient has mesothelioma in situ or as an early-stage invasive mesothelioma if radiologic presentation of typical diffuse pleural mesothelioma is absent.

The authors should mention the idea of mesothelioma in situ which has been reintroduced recently as a clinico-pathologic entity. The authors also should write that the management of patients with mesothelioma in situ has not been established and these patients should be discussed at multidisciplinary team meetings. The authors should cite the manuscripts below.

Churg A, et al. Mod Pathol. 2020 Feb;33(2):297-302.

Klebe S, et al. Pathology. 2021 Mar 25:S0031-3025(21)00062-3.

Page 17, line 3. Heterogeneity of mesothelioma

Figures 15 are images of breast carcinoma and not related to heterogeneity of mesothelioma. The authors should use the appropriate images to explain the heterogeneity of mesothelioma.

Page 17, line 7.

For the explanation of the difference of diagnosis between biopsies and surgical samples, difference in size of samples between biopsy and surgical samples matter. Pathologists can observe wide area of the mesothelioma with surgical samples but only limited area with small biopsy. This should be mentioned.

Author Response

(The authors gave the same response as above.)

Round 2

Reviewer 2 Report

The authors have revised the manuscript based on the suggestion by the reviewers.